# Sperm induction of somatic cell-cell fusion as a novel functional test

**Nicolas G Brukman\*, Clari Valansi, Benjamin Podbilewicz\***

Department of Biology, Technion-Israel Institute of Technology, Haifa, Israel

**Abstract** The fusion of mammalian gametes requires the interaction between IZUMO1 on the sperm and JUNO on the oocyte. We have recently shown that ectopic expression of mouse IZUMO1 induces cell-cell fusion and that sperm can fuse to fibroblasts expressing JUNO. Here, we found that the incubation of mouse sperm with hamster fibroblasts or human epithelial cells in culture induces the fusion between these somatic cells and the formation of syncytia, a pattern previously observed with some animal viruses. This sperm-induced cell-cell fusion requires a species-matching JUNO on both fusing cells, can be blocked by an antibody against IZUMO1, and does not rely on the synthesis of new proteins. The fusion is dependent on the sperm's fusogenic capacity, making this a reliable, fast, and simple method for predicting sperm function during the diagnosis of male infertility.

## Editor's evaluation

The molecular mechanisms of mammalian gamete fusion are still obscure. In this study, the authors provide compelling evidence that binding of sperm to somatic cells expressing egg counterpart proteins enables non-reproductive cells to efficiently undergo cell-cell fusion and form a syncytium. This is an important finding that establishes a new experimental system to investigate the proteins involved in egg-sperm binding, and could accelerate innovation of male infertility diagnostic assays.

**\*For correspondence:**
nbrukman@gmail.com (NGB);
podbilew@technion.ac.il (BP)

## Introduction

Infertility is estimated to affect approximately 15% of the population (*World Health Organization, 2023*), and many couples dealing with this burden turn to assisted reproductive techniques (ARTs) as a possible solution (*Steptoe and Edwards, 1978*; *Palermo et al., 1992*). During fertility evaluation of a male patient, a basic semen examination is performed, where different sperm parameters are determined: concentration, motility, morphology, and vitality (*World Health Organization HRP, 2021*). This basic analysis can be complemented with additional tests to determine the fertilizing potential of the sperm and choose the best treatment for the couple. The hamster oocyte penetration (HOP) test has been proposed as a quantitative method for analyzing the fusogenic potential of human spermatozoa (*Aitken and Elton, 1986*; *Yanagimachi et al., 1976*; *Zainul Rashid et al., 1998*), however, it was excluded from the newest WHO laboratory manual for the examination and processing of human semen (*World Health Organization HRP, 2021*) for being considered obsolete. Therefore, to date, there is no other standardized methodology to analyze specifically the ability of sperm to fuse to oocytes.

 In mammals, the adhesion of the sperm to the oocyte plasma membranes is mediated by the species-specific interaction of two membrane proteins: IZUMO1 and JUNO (*Bianchi and Wright, 2015*; *Aydin et al., 2016*; *Ohto et al., 2016*; *Matsumura et al., 2021*). The transmembrane protein IZUMO1 is expressed during spermatogenesis and localizes to the fusogenic region of the sperm head after an exocytic process named acrosome reaction (*Inoue et al., 2005*; *Satouh et al., 2012*; *Inoue et al., 2015*; *Nishimura et al., 2016*). On the other hand, the IZUMO1 receptor, JUNO, is an oocyte protein bound to the plasma membrane by a GPI lipid anchor (*Bianchi et al., 2014*). The subsequent fusion of

the two gametes relies on the action of IZUMO1 in a unilateral manner (*Brukman et al., 2023*). The IZUMO1-JUNO interaction was characterized also in humans (*Aydin et al., 2016*; *Ohto et al., 2016*) and its relevance for human fertility is supported by the presence of antibodies against IZUMO1 in the sera of immuno-infertile women (*Clark and Naz, 2013*; *Yu et al., 2018*; *Enoiu et al., 2022*), the mutations on JUNO in cases of fertilization failure and polyspermy (*Clark and Naz, 2013*; *Yu et al., 2018*; *Enoiu et al., 2022*), and the lower levels of IZUMO1 observed in sperm that failed to fertilize during in vitro fertilization (IVF) treatments (*Clark and Naz, 2013*; *Yu et al., 2018*; *Enoiu et al., 2022*).

Recently, we found that mouse sperm can fuse to fibroblasts ectopically expressing the sperm-receptor JUNO (*Brukman et al., 2023*). Here, we show that after sperm fuses with a somatic cell, this cell can fuse with additional cells inducing syncytia formation - a single cell with several nuclei (multinucleated cell). This is possibly mediated by the bridging of a single sperm simultaneously fused to two different cells. We call this process 'Sperm-induced cell-cell fusion requiring JUNO (SPICER).' We found that sperm with higher fertilizing ability can induce the fusion of somatic cells more efficiently, judged by the increased levels of multinucleation. This establishes the basis for the future development of a new method for diagnosis of male fertility hinge on the ability of sperm to induce cell-cell fusion in vitro.

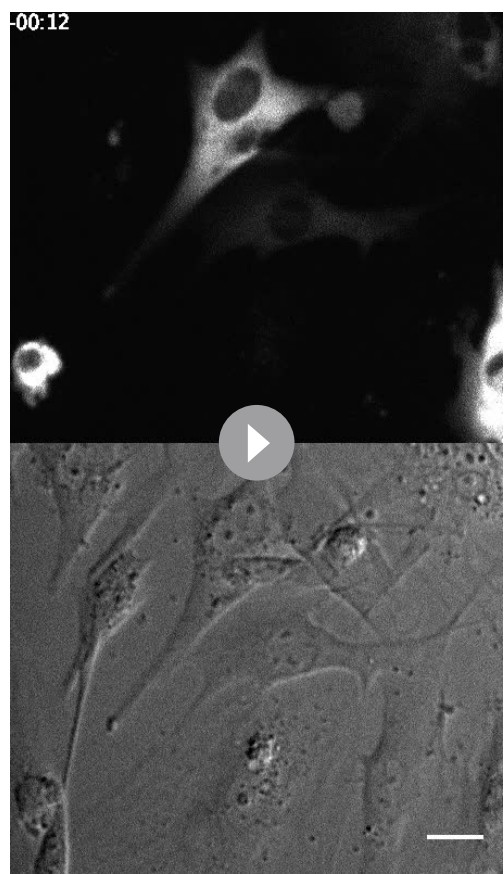

**Video 1.** Time-lapse experiment using spinning disk confocal microscopy showing the fusion of two cells expressing JUNO and GFPnes after the addition of sperm at t=0 min. Time in hours:minutes. Green and DIC channels are shown. Scale bars, 20 μm.
https://elifesciences.org/articles/94228/figures#video1

## Results

### Sperm fusion to fibroblasts promotes syncytia formation

There are reports showing that mammalian sperm can fuse to somatic cells (*Rival et al., 2019*; *Mattioli et al., 2009*; *Bendich et al., 1974*). In our experimental conditions, mouse sperm cells only fuse to Baby Hamster Kidney (BHK) cells if they are induced to express the sperm-receptor, JUNO (*Brukman et al., 2023*). This fusion was demonstrated by detecting the transfer of the DNA binding GFP (GFP-MBD) from the BHK cells to the sperm heads (*Brukman et al., 2023*). To further study the mechanisms of mammalian sperm-oocyte fusion, we incubated mouse sperm with BHK cells expressing JUNO (functioning as pseudo-oocytes) and determined the efficiency of the sperm-BHK cell interactions. Surprisingly, we found that sperm induces the formation of multinucleated BHK cells (syncytia; *Figure 1A–C*). The induction of multinucleation was dependent on the presence of JUNO (*Figure 1B–C*), however, JUNO expression alone is not sufficient to induce this process. Only cells with sperm fused to them form syncytia (*Figure 1C*). Furthermore, bigger syncytia tend to contain more sperm (*Figure 1—figure supplement 1A*) and the levels of multinucleation were dependent on the amount of sperm added to the cells (*Figure 1—figure supplement 1B*). Thus, the fusion of sperm with JUNO-expressing BHK cells is required for inducing subsequent multinucleation of these somatic cells. This phenomenon was unexpected, leading us to further study the mechanisms of this process.

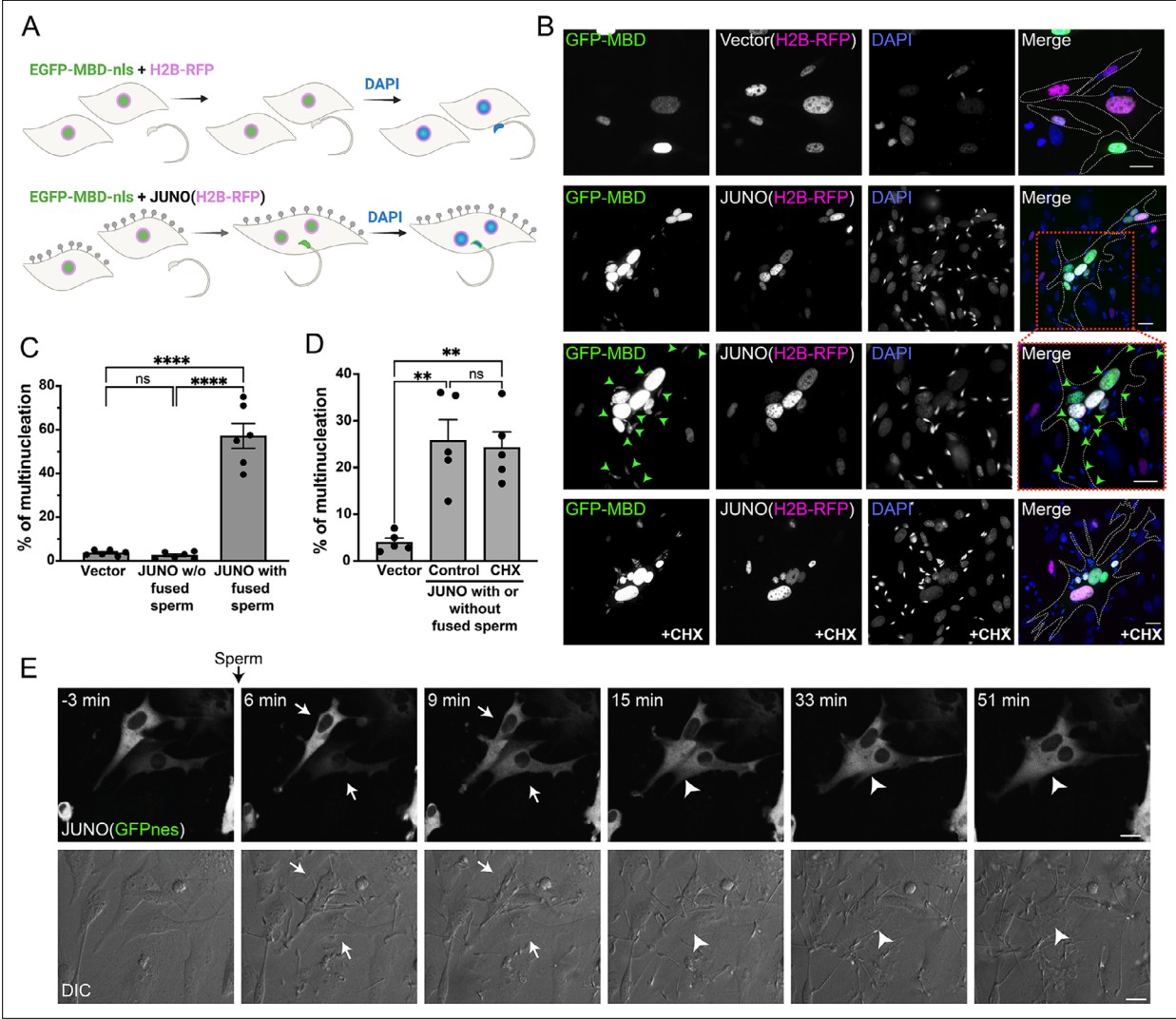

**Figure 1.** Sperm induce syncytia formation of fibroblasts. (**A**) Scheme of experimental design: Baby hamster kidney (BHK) cells were transfected with either pCI::H2B-RFP or pCI::JUNO::H2B-RFP vectors and pcDNA3.1-EGFP-MBD-nls. Mouse sperm were obtained from the epididymis of adult mice and capacitated in HTF capacitating medium. Sperm cells were co-incubated with the BHK cells for 4 hr and then the cells were fixed, and stained with DAPI to detect the DNA. Gray lollipops represent JUNO molecules. (**B**) Representative images showing H2B-RFP (magenta), GFP-MBD (green), DAPI (blue) channels, and the merge. Dotted lines contour relevant cells. Green arrowheads point to fused sperm (GFP-positive). The presence of 200 μg/ml of cycloheximide is indicated (CHX). Scale bars, 20 μm. (**C**) The percentage of multinucleation was defined as the ratio between the nuclei in multinucleated cells (NuM) and the total number of nuclei in fluorescent cells (NuF), as follows: % of multinucleation = (NuM/NuF) × 100. We show individual data and means ± SEM of six independent experiments. The number of nuclei counted per experiment and per treatment was 500. For JUNO-transfected cells, multinucleation was counted separately for cells with and without sperm fused to them. Comparisons were made with one-way ANOVA followed by Tukey's test. ns = non-significant, ****$p < 0.0001$. (**D**) In another set of experiments, JUNO-transfected cells were treated with 200 μg/ml CXH to inhibit de novo synthesis of proteins. Multinucleation was quantified for the whole population of transfected cells. We show individual data and means ± SEM of five independent experiments. ns = non-significant, **$p < 0.01$. (**E**) Time-lapse images from a movie showing sperm-induced cell-cell fusion. BHK cells were transfected with the pCI::JUNO::GFPnes plasmid and sperm were added at time = 0 min. Arrows and arrowheads indicate contacting and fused cells, respectively. The green channel (GFPnes) and the DIC images are shown (see also *Video 1*). Scale bars, 20 μm.

The online version of this article includes the following figure supplement(s) for figure 1:

**Figure supplement 1.** Effect of sperm number on syncytia formation.

**Figure supplement 2.** IZUMO1 diffuses into the target cell upon sperm fusion.

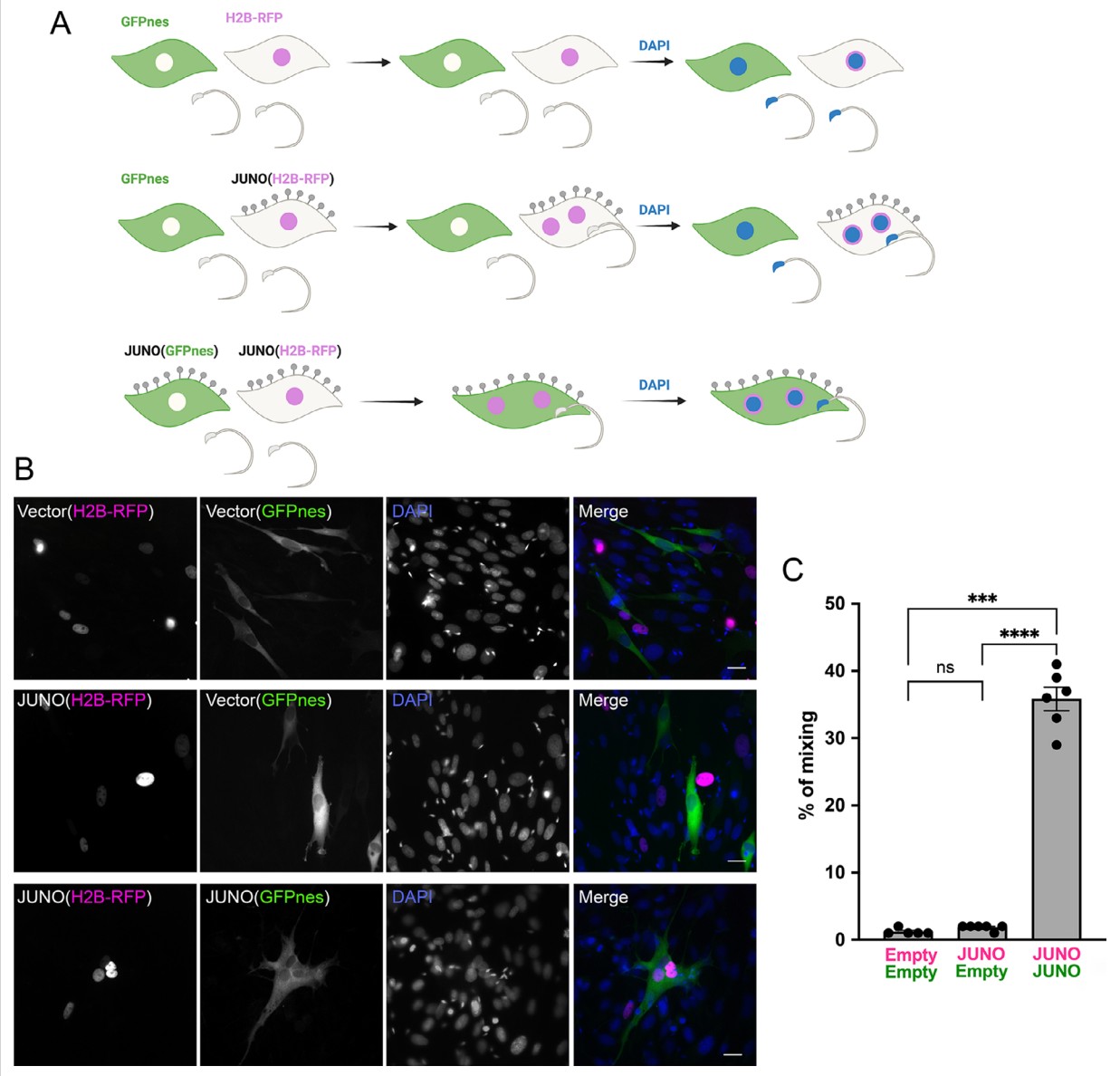

**Figure 2.** Sperm-induced fusion is dependent on JUNO. (**A**) Scheme of experimental design. Baby hamster kidney (BHK) cells transfected with pCI::GFPnes or pCI::H2B-RFP empty vectors or containing the coding sequence for the expression of JUNO were mixed as indicated. Later, the cells were co-incubated with sperm for 4 hr, fixed, and stained with DAPI. (**B**) Representative images for each treatment. Each separate channel for GFPnes (green cytoplasm), H2B-RFP (magenta nuclei), and DAPI staining (blue) are shown. Scale bars, 20 μm. Fused, mixed cells contain both GFPnes and H2B-RFP staining. (**C**) Quantification of content mixing experiments. The percentage of mixing was defined as the ratio between the nuclei in mixed cells (NuM) and the total number of nuclei in mixed cells and fluorescent cells whose cell bodies are in contact that did not fuse (NuC), as follows: % of mixing = (NuM/[NuM +NuC]) × 100. Bar chart showing individual experiment values (each corresponding to 1000 nuclei) and means ± SEM of six independent experiments. Comparisons by one-way ANOVA followed by Tukey's test. ns = non-significant, ***p<0.001, ****p<0.0001.

The online version of this article includes the following figure supplement(s) for figure 2:

**Figure supplement 1.** The timing of sperm fusion to somatic cells is critical for content mixing.

## Sperm induce syncytia formation using a viral-like mechanism

To our knowledge, sperm-induced cell-cell fusion has not been described in any species. However, decades ago, it was first described that some somatic cells can fuse following viral infection (*Okada, 1962*; *Kohn, 1965*) and later confirmed for diverse viruses, including SARS-CoV2 (*Buchrieser et al.,*

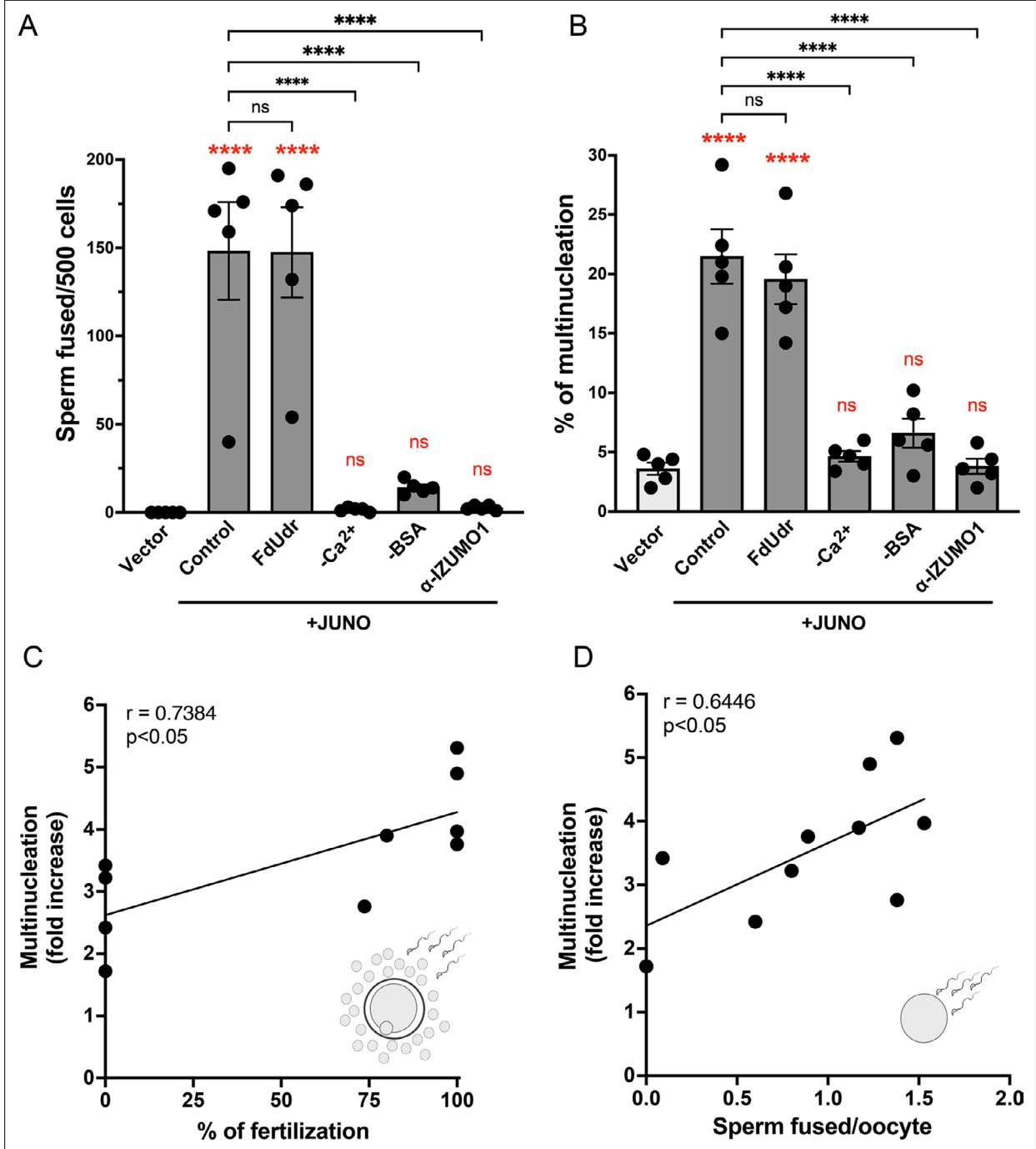

**Figure 3.** Syncytia formation requires a functional sperm, is dependent on IZUMO1 and is not affected by an inhibitor of cell division. Baby hamster kidney (BHK) cells were transfected with pcDNA3.1-EGFP-MBD-nls together with pCI::H2B-RFP (empty vector) or pCI::JUNO::H2B-RFP and co-incubated with control sperm in the presence of 20 µM of the inhibitor of cell division FdUdr or of 1 µg/µl of anti-IZUMO1 antibody (clone Mab120). Alternatively, cells were mixed with sperm incubated in a medium lacking calcium (-Ca$^{2+}$) or bovine serum albumin (-BSA); both conditions fail to support sperm capacitation. The number of sperm fused per 500 fluorescent cells (**A**) and the percentage of multinucleation (**B**) were determined. Bar charts showing individual experiment values and means ± SEM of five independent experiments. Comparisons by one-way ANOVA followed by Tukey's test. In red are the comparisons against the empty vector. ns = non-significant, ***p<0.001, ****p<0.0001. (**C–D**) Multinucleation levels are relative to the control without sperm as a function of the percentage of fertilized eggs when cumulus-oocytes complexes were used (**C**), or as a function of the number of sperm fused per oocytes when ZP-free eggs were employed (**D**). Each dot corresponds to a different mouse. The Pearson's coefficient 'r' and the significance are included in each panel.

*Figure 3 continued on next page*

*Figure 3 continued*

The online version of this article includes the following figure supplement(s) for figure 3:

**Figure supplement 1.** Acrosome reaction is reduced in non-capacitated sperm, but does not correlate with sperm-induced cell-cell fusion requiring JUNO (SPICER) levels in capacitated conditions.

---

*2020*). This process may require the synthesis of new viral proteins and, therefore, induces Fusion From Within (FFWI) while other viruses induce fusion independently of protein synthesis in a process called Fusion From Without (FFWO) (*Bratt and Gallaher, 1969*; *Duelli and Lazebnik, 2007*). For instance, orthoreoviruses induce the expression of fusion-associated small transmembrane (FAST) proteins upon infection, promoting FFWI (*Ciechonska et al., 2014*; *Theuerkauf et al., 2021*). We hypothesized that in a way reminiscent of viral-induced cell-cell fusion, sperm could induce syncytia formation following its merger via FFWI or FFWO. To distinguish between these alternative mechanisms, we performed the experiment in the presence of cycloheximide to inhibit protein synthesis. We found no differences in the levels of multinucleation between control and cycloheximide treatments (*Figure 1B and D*). This shows that de novo protein synthesis is not required for the induction of multinucleation by sperm and suggests a mechanism of FFWO. Additionally, live imaging of the process showed that fusion between cells occurs efficiently within 10–15 min after the addition of the sperm (*Figure 1E*, *Video 1*), suggesting that protein synthesis is not involved. Even though it is not known whether sperm can induce the fusion of somatic cells in vivo, we show that sperm use a viral-like mechanism of FFWO when the fibroblasts express the oocyte JUNO in vitro.

## Syncytia formation results from the fusion of cells expressing JUNO

Then we asked whether the sperm-induced multinucleation was a consequence of BHK-BHK cell fusion and if so, whether it requires JUNO to be present on both fusing cells. For that, we employed a content-mixing experiment where two populations of cells expressing different fluorescent markers were mixed and then exposed to sperm (*Figure 2A*). We observed the formation of multinucleated hybrid cells containing both fluorescent markers, confirming the fusion of BHK cells expressing JUNO (*Figure 2*). This BHK-BHK cell fusion was not observed when only one or neither of the cell populations expressed JUNO (*Figure 2B–C*), indicating that sperm-induced multinucleation is indeed BHK-BHK cell fusion which relies on bilateral expression of JUNO. An alternative explanation for sperm-induced multinucleation is a failure in cytokinesis. However, the presence of an inhibitor of the cell cycle (FdUdr, 5-fluoro-2′-deoxyuridine, *Valansi et al., 2017*) does not inhibit multinucleation (*Figure 3A–B*), ruling out that syncytia formation is as a consequence of failure in the division of the BHK cells and confirming the occurrence of fusion between them. Following these findings, we named this process SPICER that stands for 'SPerm-Induced CEll-cell fusion Requiring JUNO'.

## Sperm fuses simultaneously with somatic cells via a sandwich mechanism

Two models may explain FFWO induced by sperm (*Figure 1—figure supplement 2C*), similar to what was reported for viruses (*Tang et al., 2021*). The first model involves an initial fusion of the sperm to a BHK cell expressing JUNO, the transfer of IZUMO1 to the target cell (forming a pseudo-sperm from a pseudo-oocyte), and a subsequent fusion to a new JUNO-expressing BHK (*Figure 1—figure supplement 2Ci*). On the other hand, the second model suggests the fusion of one sperm cell with two JUNO-positive BHK cells, bridging both and forming a continuous syncytium (*Figure 1—figure supplement 2Cii*). We have previously shown that IZUMO1 fades from the sperm after fusion to the fibroblast (*Brukman et al., 2023*) indicating that this transmembrane protein is transferred to the target cell plasma membrane. Consistent with these findings, IZUMO1 was observed diffusing to the oocyte plasma membrane during in vitro fertilization assays (*Satouh et al., 2012*). Here, we detected IZUMO1 by immunostaining and by a fluorescent reporter and confirmed that it diffuses from the sperm head after its fusion with the BHK cell (*Figure 1—figure supplement 2A–B*), suggesting a function for the fusogenic machinery carried by the sperm during the FFWO process. However, when sperm were first allowed to fuse with one population of green cells before adding the second

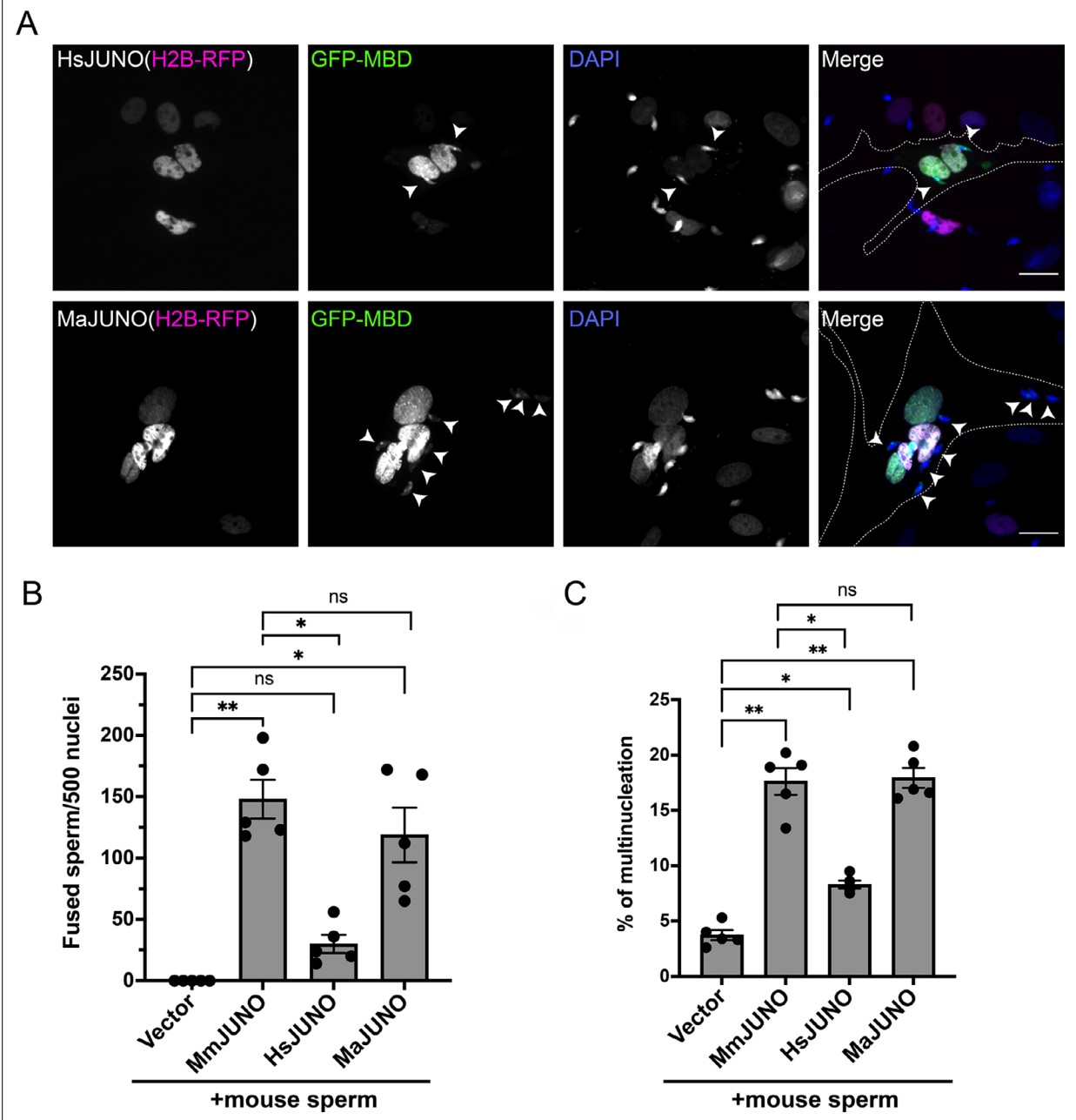

**Figure 4.** Sperm induce BHK-BHK cell fusion in a species-specific manner. Baby hamster kidney (BHK) cells were transfected with pcDNA3.1-EGFP-MBD-nls together pCI::H2B-RFP (empty vector) or pCI::JUNO::H2B-RFP encoding for human JUNO (HsJUNO), mouse JUNO (MmJUNO) or hamster JUNO (MaJUNO), and co-incubated with capacitated mouse sperm. (**A**) Representative images showing H2B-RFP (magenta), GFP-MBD (green), DAPI (blue) channels, and the merge. Dotted lines contour a cell transfected with human or hamster JUNO with mouse sperm fused (arrowheads). Scale bar, 20 µm. The number of sperm fused per 500 fluorescent cells (**B**) and the percentage of multinucleation (**C**) were determined. Bar charts showing individual experiment values and means ± SEM of five independent experiments. Comparisons by one-way ANOVA followed by Tukey's test. *p<0.05, **p<0.01, ***p<0.001.

population of red cells, hybrid syncytia were not observed (*Figure 2—figure supplement 1A–B*). Only when a viral fusogen (VSV-G) was employed or when the two populations of cells were plated before the addition of the sperm, content mixing was induced. These results suggest that it is not possible to uncouple temporarily the sperm-BHK and BHK-BHK cell fusions, arguing against the model of transfer of fusion proteins and suggesting that sperm are fusing simultaneously to two cells using a sperm sandwich mechanism (*Figure 1—figure supplement 2Cii*).

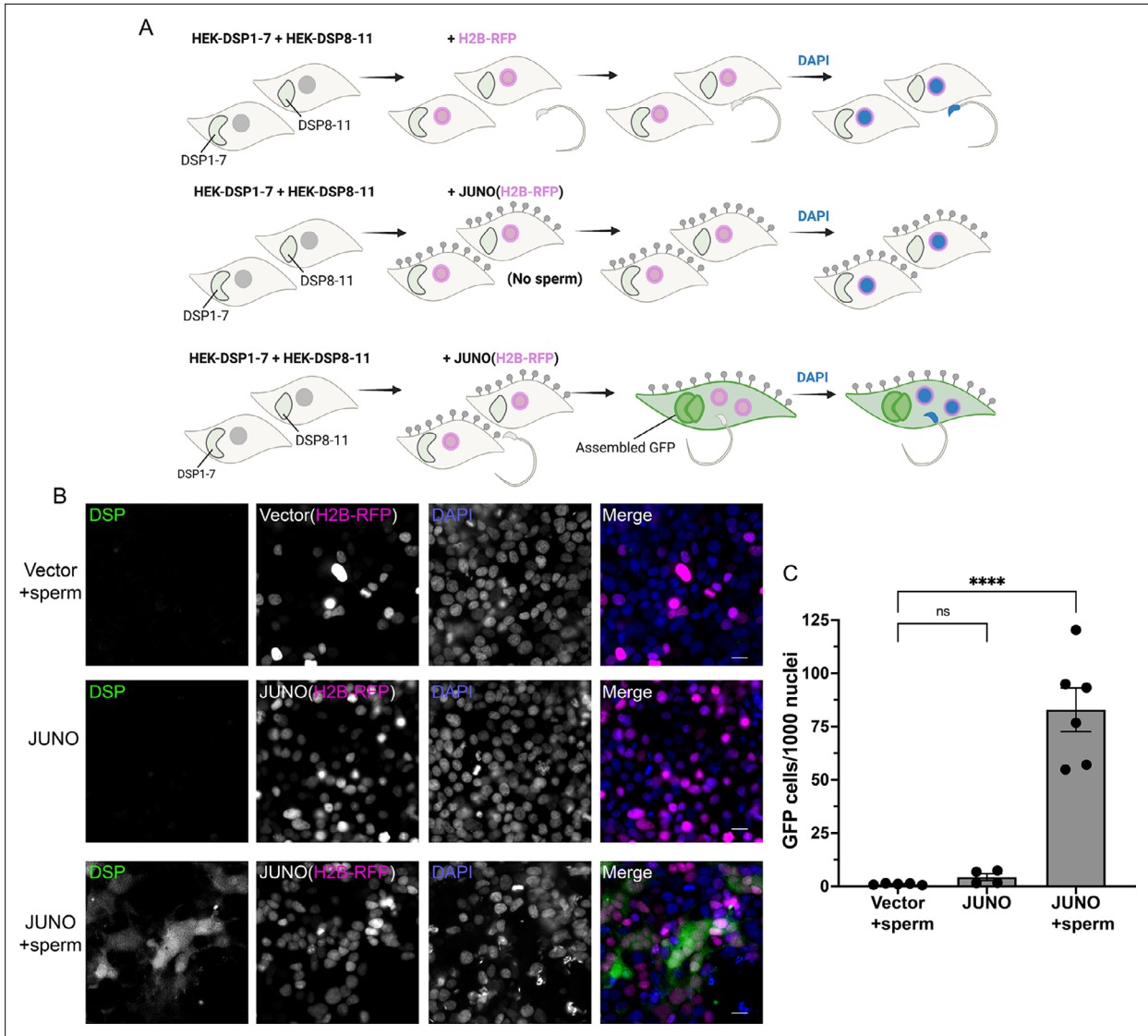

**Figure 5.** Sperm-induced fusion of human cells evaluated by dual split proteins (DSP). (**A**) Scheme of experimental design. HEK293T cells stably expressing the split GFP (DSP1-7 or DSP8-11) were mixed and transfected with pCI::H2B-RFP or pCI::JUNO::H2B-RFP vectors. When indicated, the cells were later co-incubated with sperm for 4 hr, fixed, and stained with DAPI. (**B**) Representative images for each treatment. Each separate channel for GFP (assembled GFP in fused cells), H2B-RFP (magenta nuclei), and DAPI staining (blue) are shown. Scale bars, 20 µm. (**C**) Quantification of content mixing experiments. The extent of fusion was determined by counting the number of GFP-positive cells per 1000 nuclei. Bar chart showing individual experiment values (each corresponding to 1000–2000 nuclei) and means ± SEM of at least four independent experiments. Comparisons by one-way ANOVA followed by Tukey's test. ns = non-significant, ****p<0.0001.

The online version of this article includes the following figure supplement(s) for figure 5:

**Figure supplement 1.** Time course of sperm-induced fusion of human cells evaluated by dual split proteins (DSP).

## Sperm-induced multinucleation as a readout of sperm fertilizing ability

Following the discovery of the ability of sperm to induce cell-cell fusion, we decided to evaluate whether BHK cell multinucleation could be used as a readout of sperm fusogenic potential. For this purpose, we incubated the sperm in media lacking BSA or $Ca^{2+}$ that do not support capacitation (i.e. a process by which sperm acquires its fusogenic activity) (*Figure 3—figure supplement 1*; *Visconti et al., 1995*; *Yanagimachi, 1988*). We found that sperm cells incubated under these conditions failed to fuse to BHK cells, as well as to induce syncytia formation (*Figure 3A–B*). Thus, BHK-BHK cell fusion

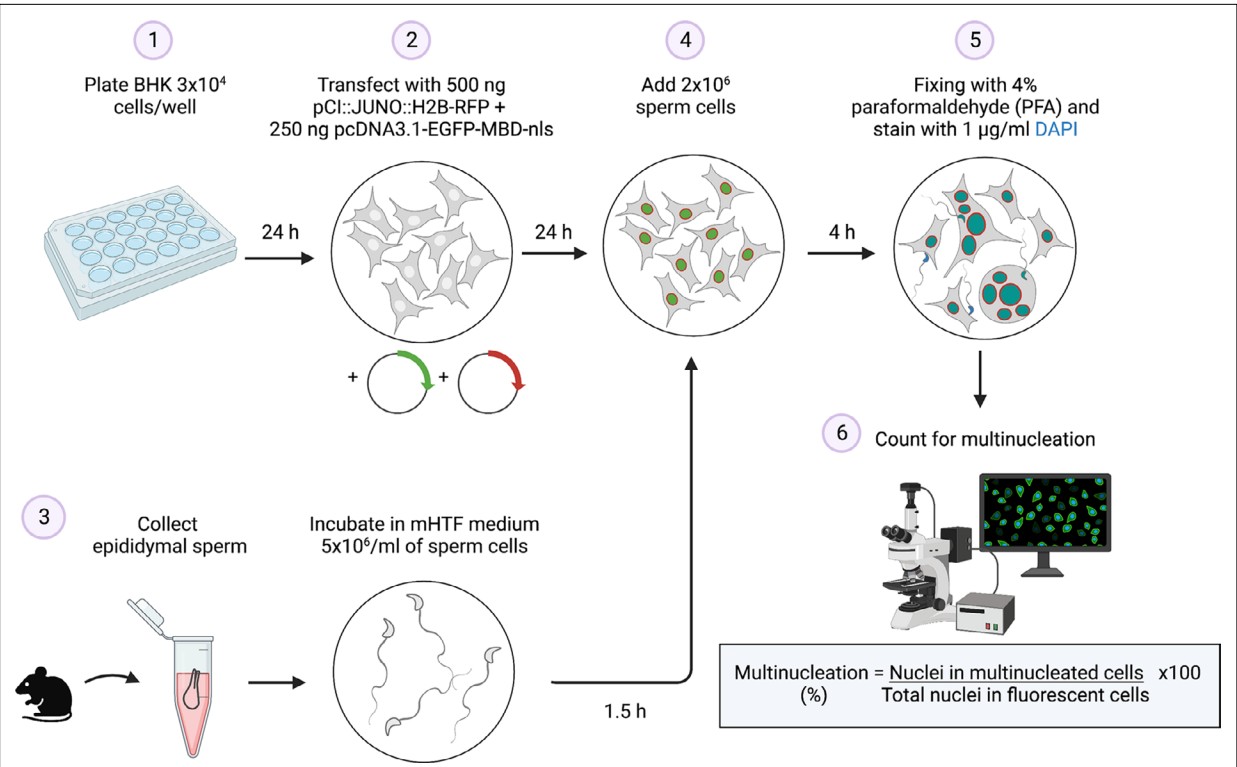

**Figure 6.** Summary of the sperm-induced cell-cell fusion requiring JUNO (SPICER) method. Schematic representation of the multinucleation assay employed to determine the sperm fusogenic potential. (1) Baby hamster kidney (BHK) cells are seeded on a plate. (2) The cells are transfected with the plasmids encoding for JUNO and EGFP-MBD. (3) Mouse sperm cells are collected and capacitated. (4) The BHK and sperm cells are co-incubated. (5) The cells are washed, fixed, and stained with DAPI. (6) Multinucleation levels are quantified as indicated.

requires fully capacitated sperm. A sperm which is unable to fuse with BHK cells, will not induce multi-nucleation. Furthermore, to examine whether the extent of cell-cell fusion correlates with the sperm fertilizing ability, we simultaneously evaluated the levels of multinucleation and the performance of sperm during in vitro fertilization assays. We detected a positive and significant correlation between the syncytia formation and the levels of fertilization, evaluated with complete and denuded oocytes (*Figure 3C–D*, *Source data 1*). In contrast, BHK multinucleation did not correlate with the percentage of acrosome reaction of capacitated sperm (*Figure 3—figure supplement 1B*), suggesting that SPICER relies not only on capacitation but on the overall sperm fertilizing potential. Together, these results support the use of this assay as a predictor of sperm fertilizing ability.

## SPICER depends on the activity of IZUMO1

The sperm IZUMO1 protein is essential for sperm-egg interactions via binding to JUNO (*Bianchi et al., 2014*). More recently, IZUMO1 was shown to induce the fusion of cells in a JUNO-independent way using a different domain that is not required for binding to the egg receptor (*Brukman et al., 2023*). To study whether IZUMO1 is required for the fusion of sperm to BHK cells and for the induction of multinucleation of the fibroblasts, we decided to inhibit IZUMO1 activity using a blocking antibody. For this purpose, we used a monoclonal antibody against IZUMO1 which has been shown to inhibit sperm-egg fusion (*Inoue et al., 2013*) in our sperm-induced multinucleation assay. We found that the anti-IZUMO1 antibody had the ability to block both sperm-BHK and sperm-induced BHK-BHK cell fusions even when sperm cells were fully capacitated (*Figure 3A–B*). Altogether, our results show that the SPICER assay is able to determine sperm fertilization potential that depends on the IZUMO1-JUNO interactions.

## Cell-cell fusion requires species-specific JUNO

Elegant work on IZUMO1-JUNO interactions among different mammalian species has shown species-specific selectivity between these crucial interactions (*Bianchi and Wright, 2015*). To evaluate the effect of species-specificity of the SPICER assay, we incubated the mouse sperm with BHK cells expressing human JUNO. Under these conditions, we observed lower levels of sperm-BHK cell fusion and a reduction in BHK cell multinucleation compared to cells expressing mouse JUNO (*Figure 4A–C*). Interestingly, the low activity mediated by human JUNO was still significant compared to the control without JUNO. This confirms the requirement of a species-matching JUNO for the assay; however, it puts in evidence a residual cross-interaction between mouse sperm and human JUNO. On the contrary, when hamster JUNO was employed, no differences were observed compared to mouse JUNO (*Figure 4A–C*), consistent with the promiscuous nature of the hamster oocytes (*Bianchi and Wright, 2015*; *Nakajima et al., 2022*; *Hanada and Chang, 1972*).

## Sperm induces the fusion of human epithelial cells

To exclude any BHK cell-specific requirement for SPICER, we tested the epithelial human embryonic kidney 293T (HEK) cells. Due to the smaller size and more rounded shape of these cells that hinder the quantification of multinucleation, we opted to analyze fusion by content mixing. For this, we utilized the Dual Split Protein system (*Wang et al., 2014*; *Ishikawa et al., 2012*) which employs two GFP halves expressed in different populations of cells. Upon fusion of these two populations, the reporter self-assembles and the cells become fluorescent (*Nakane and Matsuda, 2015*). The number of green fluorescent cells relative to the number of transfected cells was determined as an indication of fusion levels (*Figure 5A*). As observed for the BHK cells, HEK cells bearing JUNO fused between them when they were co-incubated with sperm (*Figure 5B–C*). On the contrary, neither HEK cells without JUNO nor HEK expressing JUNO but without sperm showed content mixing (*Figure 5B–C*). The fusion was dependent on the co-incubation time reaching a maximum at around 4 hr (*Figure 5—figure supplement 1*). These results show that SPICER is not restricted to BHK cells or fibroblasts and opens the possibility of adapting this assay to different cell lines and approaches to evaluate cell-cell fusion.

# Discussion

In this study, we described a new phenomenon in which sperm cells can induce the fusion of cells ectopically expressing JUNO in culture, resembling the viral-like fusion of cells upon infection. This process is likely mediated by the simultaneous fusion of sperm to adjacent cells and the extent of multinucleation was correlated with the sperm fertilizing potential.

## SPICER occurrence in vivo

Apart from being expressed in oocytes, JUNO has been described in lymphoid tissues (*Spiegelstein et al., 2000*), more specifically in regulatory T cells (*Yamaguchi et al., 2007*). Therefore, we cannot exclude that sperm could bind and fuse to other cell types in vivo, and whether it has any physiological or pathological relevance. This fusion cannot occur before acrosomal exocytosis, when IZUMO1 is enclosed into the acrosome (*Inoue et al., 2005*; *Satouh et al., 2012*), and therefore, it will be relevant to study sperm fusion to somatic cells within the oviduct of the female, close to the fertilization site (*Muro et al., 2016*; *Hino et al., 2016*; *La Spina et al., 2016*). Furthermore, JUNO was detected in the first polar body (*Suzuki et al., 2017*) and whether sperm can fuse to it or if sperm fusion to the oocyte can induce the polar body-egg fusion remains unknown.

## Expanding SPICER across species

The assay that we described here can be a potent tool to study cross-fertilization between different species and mechanisms of speciation and evolution, presenting a step beyond biochemical approaches to study IZUMO1-JUNO interactions (*Bianchi and Wright, 2015*). On the other hand, by simply exchanging the JUNO sequence used, this assay could be easily adapted to allow the analysis of the sperm fertilizing ability in different mammalian species, such as humans or cattle, having implications not only for reproductive health but also for biotechnological and agricultural uses. Furthermore, even though IZUMO1-JUNO interaction is a specific requirement during sperm-egg fusion of mammals (*Grayson, 2015*), we cannot exclude that alternative configurations, using the right egg receptors (*Lu*

and Ikawa, 2022), will be possible in the future to induce somatic cell-cell fusion by sperm of other sexually reproducing organisms. In species where *Izumo1* and *Izumo1r* genes are absent, such as plants and protists, the gamete fusion is promoted by the fusogen HAPLESS 2/GENERATIVE CELL-SPECIFIC 1 (HAP2/GCS1) (*Brukman et al., 2021*). In other organisms like zebrafish or nematodes, IZUMO1 orthologs were described (*Binner et al., 2021*; *Nishimura et al., 2015*; *Takayama et al., 2021*) but, since JUNO is present only in mammals (*Vance and Lee, 2020*; *Grayson, 2015*), it is unclear whether male gametes in other species interact with different receptors on the female gamete. Therefore, it can be speculated, for instance, that the expression of a species-specific Bouncer will be required to activate the fusion of fish sperm (*Gert et al., 2023*; *Herberg et al., 2018*) or *Chlamydomonas* FUS1 to mediate the fusion of the algae minus gamete (*Pinello et al., 2021*; *Ferris et al., 1996*).

## SPICER as a diagnostic tool

Determining sperm's fusogenic potential is of great interest for infertility diagnosis of both human and stud animals. Considering that the HOP assay became obsolete, an assay that evaluates the fusion of somatic cells with ectopic expression of JUNO induced by competent sperm (SPICER, *Figure 6*) represents a powerful diagnostic tool. SPICER could potentially predict the success chances of ARTs like intrauterine insemination (IUI) or conventional IVF, which require a fusion-competent sperm, and if it is necessary to proceed with a more complex technique such as intracytoplasmic sperm injection (ICSI). This is particularly important considering the routine use of the ICSI is under debate as it is associated with a slightly higher risk of adverse outcomes in the progeny (Practice Committees of the American Society for *Practice Committees of the American Society for Reproductive Medicine and the Society for Assisted Reproductive Technology, 2020*).

Considering our previous work showing that IZUMO1 can mediate cell-cell fusion (*Brukman et al., 2023*) and that significantly lower levels of this protein were detected in the sperm cells of patients with total fertilization failure (*Enoiu et al., 2022*), we expect that the SPICER assay will be able to resolve some unexplained cases of male infertility that result from loss of function of components of the fusion machinery. In this context, other proteins that are essential for gamete fusion, such as SPACA6, TMEM95, TMEM81, FIMP, SOF1, and DCST1/2 (*Barbaux et al., 2020*; *Noda et al., 2020*; *Lorenzetti et al., 2014*; *Lamas-Toranzo et al., 2020*; *Fujihara et al., 2020*; *Inoue et al., 2021*; *Noda et al., 2022*; *Elofsson et al., 2023*; *Deneke et al., 2023*), might be transferred to the somatic cells together with IZUMO1 during sperm-induced cell-cell fusion. Finally, considering that antibodies against IZUMO1 were detected on the sera of infertile women (*Clark and Naz, 2013*) and here, we observed that a monoclonal antibody against this protein was able to block sperm-induced syncytia, SPICER could serve as a tool to readily analyze cases of immuno-infertility. For all these cases, in-depth studies utilizing human sperm from healthy donors and infertile patients will be necessary.

## Further applications for SPICER

In addition to the potential uses of SPICER for the diagnosis of infertility, this assay could be used to evaluate potential sperm donors and animal studs. Moreover, our SPICER tool has the potential to aid screening for compounds that enhance fertilization (new fertility treatments) or that block gamete interactions (*Stepanenko et al., 2022*), or to easily determine the effect of genetic variations of JUNO (*Allingham and Floriano, 2021*; *Takaiso et al., 2016*). SPICER could also facilitate studies aimed at resolving standing enigmas of sperm-egg fusion (*Deneke and Pauli, 2021*; *Yanagimachi, 2022*). This includes the role of IZUMO1 and other potential fusogens as well as additional cellular pathways that are known to influence fusion in different systems, including the action of the cytoskeleton and molecular motors (*Shilagardi et al., 2013*; *Yang et al., 2017*; *Zhang et al., 2017*), specific lipids (e.g. phosphatidylserine) (*Rival et al., 2019*; *van den Eijnde et al., 2001*; *Verma et al., 2018*; *Abay et al.,*

2017), calcium signaling (*Tsuchiya et al., 2018*; *Earles et al., 2001*), phosphorylation cascades (*Eigler et al., 2021*), and more (*Brukman et al., 2019*). Finally, whether other cell types rather than sperm, such as muscle cells or osteoclasts, can induce syncytia formation after ectopic fusion remains an open question, with important biological implications.

## Limitations of the study

The main caveat of our assay is that it cannot discriminate between defects in sperm docking or gamete fusion, as a poor performance in SPICER could be explained by the altered interaction of IZUMO1 and JUNO or by a reduced capacity of IZUMO1 (and IZUMO1 partners in the sperm) to induce membrane merger. Moreover, simultaneous measurements of the levels of spontaneous acrosome reaction will be needed to exclude a faulty capacitation. Alternatively, capacitation could be further stimulated pharmacologically using a $Ca^{2+}$ ionophore (*Tateno et al., 2013*). In this context, further analysis may be required to refine the characterization of defective sperm. As noted before, future validations using human sperm will be necessary for the establishment of an assay that could be used in the clinics.

# Materials and methods

## Key resources table

| Reagent type (species) or resource | Designation | Source or reference | Identifiers | Additional information |
|---|---|---|---|---|
| Antibody | Anti-IZUMO1 Mab120 monoclonal antibody | Merck Millipore | Cat# MABT1357 | 1:500 in PBS |
| Antibody | Anti-goat IgG secondary antibody, Alexa Fluor 647 | Thermo Fisher Scientific | Cat# A-21247, RRID: AB_141778 | 1:500 in PBS |
| Strain, strain background (*Escherichia coli*) | DH5α competent cells | Thermo Fisher Scientific | Cat# 18265017 | |
| Chemical compound, drug | jetPRIME transfection reagent | Polyplus Transfection | Cat# 101000046 | |
| Chemical compound, drug | FdUdr, 5-Fluoro-2'-deoxyuridine | Sigma | Cat# F0503, CAS: 50-91-9 | |
| Chemical compound, drug | BSA, bovine serum albumin | Sigma | Cat# A7906, CAS: 9048-46-8 | |
| Chemical compound, drug | Hyaluronidase | Sigma | Cat# H3506, CAS: 37326-33-3 | |
| Chemical compound, drug | Cycloheximide | Sigma | Cat# C7698, CAS: 66-81-9 | |
| Chemical compound, drug | Puromycin Dihydrochloride | GoldBio | Cat# P-600–100, CAS: 58-58-2 | |
| Chemical compound, drug | Poly-L-lysine hydrobromide | Sigma | Cat# P2636, CAS: 25988-63-0 | |
| Cell line (*Mesocricetus auratus*) | BHK-21, clone 13 | ATCC | Cat# CCL-10, RRID: CVCL_1915 | |
| Cell line (*Homo sapiens*) | HEK293T | ATCC | Cat# CRL-3216, RRID: CVCL_0063 | |
| Cell line (*Homo sapiens*) | HEK293T-DSP1-7 | This paper | | See Materials and methods |
| Cell line (*Homo sapiens*) | HEK293T-DSP8-11 | This paper | | See Materials and methods |

*Continued on next page*

*Continued*

| Reagent type (species) or resource | Designation | Source or reference | Identifiers | Additional information |
|---|---|---|---|---|
| Strain, strain background (*Mus musculus*) | B6D2-Tg(Izumo1-mCherry) mouse line | *Satouh et al., 2012* | | |
| Strain, strain background (*Mus musculus*) | FVB/129sv/CF1 | This paper | | See Materials and methods |
| Sequence-based reagent | IZUMO1-mCherry F | *Satouh et al., 2012* | PCR primers | For genotyping: CCTTCCTGCGGCTTGTTCTCT |
| Sequence-based reagent | IZUMO1-mCherry R | *Satouh et al., 2012* | PCR primers | For genotyping:ATCAAGGTCTCAGAAC TGTTCTCCCAAACC |
| Sequence-based reagent | NheI Human Juno F | This paper | PCR primers | For cloning into pCI::H2B-RFP: TTATCGCTAGC ATGGCATGCTGGTGGCCGCTC |
| Sequence-based reagent | EcoRV Human Juno R | This paper | PCR primers | For cloning into pCI::H2B-RFP: CCAGGATATCTCAGGAAAGGAACGGCAGGAAC |
| Sequence-based reagent | NheI Hamster Juno F | This paper | PCR primers | For cloning into pCI::H2B-RFP: TTATCGCTAGCA TGGCTCAGTGGTGGCAGATTCTG |
| Sequence-based reagent | SmaI Hamster Juno R | This paper | PCR primers | For cloning into pCI::H2B-RFP: GTCCCCCGGGTCAGGAGTGGAGC AGCAGGCACAGAGAGAAGGATGT GAGGGCGTAAGAAATCTCCCGTG GAGCAGATGCGCTATTGGCG |
| Recombinant DNA reagent | pCI::H2B-RFP | *Williams et al., 2018* | RRID: Addgene #92398 | |
| Recombinant DNA reagent | pCI::GFPnes | *Moi et al., 2022* | | |
| Recombinant DNA reagent | pCI::mJUNO::H2B-RFP | *Brukman et al., 2023* | | |
| Recombinant DNA reagent | pCI::mJUNO::GFPnes | *Brukman et al., 2023* | | |
| Recombinant DNA reagent | pcDNA3.1-EGFP-MBD-nls | *Yamagata et al., 2005* | | |
| Recombinant DNA reagent | pCMV6-humanJUNO | *Wood and Wright, 2019* | | |
| Recombinant DNA reagent | pCI::humanJUNO::H2B-RFP | This paper | | Cloned with NheI and EcoRV |
| Recombinant DNA reagent | p2988::hamsterJUNO ectodomain | *Bianchi and Wright, 2015* | | |
| Recombinant DNA reagent | pCI::hamsterJUNO::H2B-RFP | This paper | | Cloned with NheI and SmaI |
| Recombinant DNA reagent | pIRESpuro3-DSP1–7 | *Wang et al., 2014* | | |
| Recombinant DNA reagent | pIRESpuro3-DSP8-11 | *Wang et al., 2014* | | |
| Recombinant DNA reagent | pCI::VSV-G::H2B-RFP | *Moi et al., 2022* | | |
| Software, algorithm | GraphPad Prism 9 | GraphPad Prism | RRID:SCR_002798 | |
| Software, algorithm | FIJI (ImageJ 1.53 c) | Image J | RRID:SCR_002285 | |

*Continued on next page*

*Continued*

| Reagent type (species) or resource | Designation | Source or reference | Identifiers | Additional information |
|---|---|---|---|---|
| Software, algorithm | Photoshop CS6 | Adobe | RRID:SCR_014199 | |
| Software, algorithm | Illustrator CS6 | Adobe | RRID:SCR_010279 | |
| Software, algorithm | ZEN microscopy software 7.0.4.0 | ZEISS | RRID:SCR_013672 | |
| Software, algorithm | MetaMorph image analysis software 7.8.1.0 | Molecular Devices | RRID:SCR_002368 | |
| Software, algorithm | Biorender | Biorender | RRID:SCR_018361 | |

## Resource availability

### Lead contact

Further information and requests for resources and reagents should be directed to and will be fulfilled by the lead contact, Benjamin Podbilewicz (podbilew@technion.ac.il).

### Materials availability

Plasmids or cell lines generated in this study are available upon request. Signing a materials transfer agreement (MTA) may be required.

## Experimental model and subject details

### Animals

All animal studies were approved by the Committee on the Ethics of Animal Experiments of the Technion - Israel Institute of Technology (reference numbers IL0670520 and IL0420321). In this study, we used wild-type male mice from a FVB/129sv/CF1 mixed background and B6D2-Tg(Izumo1-mCherry) mouse line (*Satouh et al., 2012*). Animals were bred and housed in the Technion animal facility under specific pathogen–free conditions with ad libitum access to food and water. The primers used for genotyping Izumo1-mCherry are outlined in the Key Resources Table. In all cases, mice between 3 and 6 months old were used for the experiments.

### Cell lines and DNA transfection

BHK cells (Cat# CCL-10; ATCC, RRID: CVCL_1915) and HEK293T cells (Cat# CRL-3216; ATCC, RRID: CVCL_0063) were grown and maintained in DMEM containing 10% FBS. Cells were cultured at 37 °C in 5% $CO_2$. Plasmids were transfected into cells using 2 µl jetPRIME (PolyPlus-transfection) per µg of DNA in 100 µl of reaction buffer for every ml of medium. HEK293T cells stable lines for Dual Split Proteins (DSP) 1–7 and 8–11 were prepared by transfecting pIRESpuro3-DSP1–7 and pIRESpuro3-DSP8-11, respectively and selecting with 2 µg/ml of puromycin for 10–13 days as previously described (*Wang et al., 2014*). Cell lines tested negative for mycoplasma contamination.

## Method details

### Sperm collection and capacitation

Sperm were recovered by incising the cauda epididymis, obtained from adult male mice, in 300 µl of mHTF medium (*Kito et al., 2004*) supplemented with 4 mg/ml of BSA. The sperm were diluted in fresh medium to a concentration of $5 \times 10^6$ cells/ml and incubated for 90 min at 37 °C and 5% $CO_2$ to induce capacitation.

### Sperm-to-BHK cell fusion and multinucleation

BHK cells were grown on 24-well glass bottom tissue-culture plates. 24 hr after plating, cells were transfected with 0.25 µg pcDNA3.1-EGFP-MBD-nls plasmids and 0.5 µg of either pCI::H2B-RFP or pCI::JUNO::H2B-RFP. 24 hr after transfection, $2 \times 10^6$ capacitated wild-type sperm cells in mHTF were

added to each well and co-incubated with the BHK cells for 4 hr at 37 °C and 5% $CO_2$. After one wash with PBS, the cells were fixed with 4% PFA in PBS and stained with 1 µg/ml DAPI. Micrographs were obtained using wide-field illumination using an ELYRA system S.1 microscope (Plan-Apochromat 20x NA 0.8; Zeiss). Multinucleation percentage was determined as the ratio between the number of nuclei in multinucleated cells (NuM) and the total number of nuclei in fluorescent cells (NuF), as follows: % of multinucleation = (NuM/NuF)×100. 500 nuclei (NuF) were counted in each independent repetition (experimental point). In some cases, the number of sperm fused was determined by evaluating the transfer of EGFP-MBD-nls signal from the BHK cell to the sperm nuclei (number of fused sperm/500 BHK cells independently of the amount of nuclei within it). In some experiments, sperm cells obtained from transgenic mice expressing IZUMO1-mCherry were employed to analyze IZUMO1 localization.

## In vitro fertilization

Ovulated oocytes were obtained from females previously treated with an i.p. injection of pregnant mare serum gonadotropin (5 IU; #HOR-272, Prospec), followed by an i.p. injection of human chorionic gonadotropin (5 IU, #CG5; Sigma-Aldrich) 48 hr later. Cumulus–oocyte complexes (COCs) were collected from the ampullae of induced females 12–15 hr after human chorionic gonadotropin administration in mHTF medium. COCs were inseminated with $5 \times 10^3$ capacitated sperm and co-incubated in capacitation media for 3 hr at 37 °C and 5% $CO_2$. Then, the oocytes were washed, stained with 10 µg/ml Hoechst 33342 (Sigma), and observed using wide-field illumination using an ELYRA system S.1 microscope (Plan-Apochromat 20x NA 0.8; Zeiss). Eggs were considered fertilized when at least one decondensing sperm nucleus or two pronuclei were observed in the egg cytoplasm. For fusion quantification, oocytes were denuded from the cumulus and the ZP by sequential treatment with 0.3 mg/ml hyaluronidase (H3506; Sigma-Aldrich) and acid Tyrode solution (pH 2.5; *Nicolson et al., 1975*). ZP-free eggs were inseminated with $10^3$ capacitated sperm and co-incubated for 1 hr. Then, the eggs were processed as above and the number of decondensing sperm nuclei per oocyte was scored.

## Content mixing assay using different colors

BHK cells at 70% confluence in 35 mm plates were transfected with 1 µg pCI::H2B-RFP, pCI::GFPnes, pCI::JUNO::H2B-RFP or pCI::JUNO::GFPnes. 4 hr after transfection, the cells were washed four times with DMEM with 10% serum, four times with PBS, and detached using Trypsin (Biological Industries). The cells were collected, resuspended in DMEM with 10% serum, and counted. Equal amount of H2B-RFP and GFPnes cells ($1.25 \times 10^5$ each) were mixed and seeded on glass-bottom plates (12-well black, glass-bottom #1.5 H; Cellvis) and incubated at 37 °C and 5% $CO_2$. 18 h after mixing, $4 \times 10^6$ capacitated wild-type sperm cells in mHTF were added to the BHK cells and co-incubated for 4 hr after which they were washed with PBS, fixed with 4% PFA in PBS and stained with 1 µg/ml DAPI. Micrographs were obtained using wide-field illumination using an ELYRA system S.1 microscope (Plan-Apochromat 20×NA 0.8; Zeiss). The percentage of mixing was defined as the ratio between the nuclei in mixed cells (NuM) and the total number of nuclei in mixed cells and fluorescent cells whose cell bodies are in contact that did not fuse (NuC), as follows: % of mixing = (NuM/[NuM +NuC])×100. 1000 nuclei (NuM +NuC) were counted in each independent repetition (experimental point).

## Content mixing assay in two steps

BHK cells grown on 12-well glass bottom tissue-culture plates were transfected with 1 µg of either pCI::GFPnes or pCI::JUNO::GFPnes. 24 hr later, the cells were thoroughly washed with DMEM with 10% serum, and, when indicated, $4 \times 10^6$ capacitated wild-type sperm cells in mHTF were added to the BHK cells, and co-incubated for 1 hr. In some cases, the sperm were removed by washing three times with DMEM. In parallel, BHK cells grown in 35 mm plates that were transfected with 1 µg pCI::JUNO::H2B-RFP or pCI::VSV-G::H2B-RFP the day before were washed and detached using 0.05% EDTA solution. These cells were added in a 1:1 ratio to the BHK cells previously incubated or not with the sperm, as indicated. 18 hr later, the cells were fixed with 4% PFA in PBS, stained with 1 µg/ml DAPI, and content mixing was evaluated as explained above. For activating VSV-G activity, 1 hr before fixing a 5 min incubation at pH5.5 buffer was performed (*Moi et al., 2022*).

## Content mixing assay using the dual split protein

Equal amount of HEK293T cells stably expressing DSP1-7 or DSP8-11 were mixed ($1.25 \times 10^5$ each) and seeded on glass-bottom plates (12-well black, glass-bottom #1.5 H; Cellvis) pre-treated with 20 µg/ml of Poly-L-lysine. 24 hr later, the cells were transfected with 1 µg pCI::H2B-RFP or pCI::JUNO::H2B-RFP. 18 h after transfection, $4 \times 10^6$ capacitated wild-type sperm cells in mHTF were added to the HEK293T cells and co-incubated for 4 hr after which they were washed with PBS, fixed with 4% PFA in PBS and stained with 1 µg/ml DAPI. In addition, a time course experiment was conducted where the cells were fixed at different time points. Micrographs were obtained as above, using wide-field illumination using an ELYRA system S.1 microscope (Plan-Apochromat 20x NA 0.8; Zeiss). The number of GFP-positive cells per 1000 nuclei was determined. Between 1000–2000 red nuclei were counted in each independent repetition (experimental point).

## Inhibition of sperm-induced cell-cell fusion

Different conditions were tested to evaluate their effect on sperm-induced cell-cell fusion. Non-capacitated sperm were incubated in medium mHTF lacking $Ca^{2+}$ or BSA for 90 min at 37 °C and 5% $CO_2$. In other cases, capacitated sperm were mixed before their addition to the BHK cells with 1 µg/µl of anti-IZUMO1 antibody (*Inoue et al., 2013*), 20 µM of the cell cycle inhibitor FdUdr (*Valansi et al., 2017*), or 200 µg/ml of the inhibitor of protein synthesis cycloheximide (*Wengler, 1975*).

## Evaluation of acrosome reaction

The extent of the acrosome reaction was evaluated by Coomassie brilliant blue staining as previously described (*Busso et al., 2007*). Briefly, the sperm cells were fixed in 4% paraformaldehyde in PBS for 15 min at room temperature, washed with 0.1 M ammonium acetate (pH 9) by centrifugation, mounted on slides, and air dried. Slides were successively immersed 5 min in water, 5 min in ice-cold methanol, 5 min in water, and 2 min in 0.22% Coomassie brilliant blue solution (50% methanol and 10% acetic acid). After washing with water, the samples were mounted and observed under a light microscope (X200). Sperm were scored as acrosome-intact when a bright blue labeling was observed in the dorsal region of the head or as acrosome-reacted when no staining was observed. For each condition, 1000 sperm were counted.

## Time-lapse imaging of sperm-induced cell-cell fusion

BHK cells were grown on 35 mm glass bottom tissue-culture plates (Greiner Bio-one) and, 24 hr after plating, cells were transfected with 1.5 µg pCI::JUNO::GFPnes. 24 hr after transfection, time-lapse images of the cells were imaged before and after adding $5 \times 10^6$ capacitated wild-type sperm cells in mHTF. Images of the cells were acquired every 3 min for 1 hr to record cell-to-cell fusion, using a spinning disk confocal microscope (CSU-X; Yokogawa Electric Corporation) with an Eclipse Ti inverted microscope and a Plan-Apochromat 20x (NA, 0.75; Nikon) objective. Images were obtained using an iXon3 EMCCD camera (ANDOR) through MetaMorph (Molecular Devices, version 7.8.1.0). Images in differential interference contrast and green channels were recorded.

## Immunofluorescence

The localization of IZUMO1 was determined by immunostaining. Briefly, after fixation cells were permeabilized with 0.1% Triton X-100 in PBS and incubated with anti-IZUMO1, clone Mab120 (1:500, Cat# MABT1357; Merck Millipore) followed by the secondary antibody Alexa Fluor 647 goat anti-rat (1:500, Cat# A-21247; Thermo Fisher Scientific, RRID: AB_141778). Later, the nuclei were stained with 1 µg/ml DAPI and micrographs were obtained using wide-field illumination using an ELYRA system S.1 microscope (Plan-Apochromat 20x NA 0.8; Zeiss) with an EMCCD iXon camera (Andor) through ZEN microscopy software 7.0.4.0 (RRID: SCR_013672; Zeiss).

## Quantification and statistical analysis

### Statistics and data analysis

Results are shown as means ± SEM. For each experiment, at least three independent biological repetitions were performed. The significance of differences between the averages were analyzed using one-way ANOVA, and Pearson's analysis was used to assess the correlations, as described in the legends

(GraphPad Prism 9, RRID: SCR_002798). Figures were prepared with Photoshop CS6 (Adobe, RRID: SCR_014199), Illustrator CS6 (Adobe, RRID: SCR_010279), BioRender.com (RRID: SCR_018361), and FIJI (ImageJ 1.53 c, RRID: SCR_002285).

## Acknowledgements

We thank Gavin Wright and Enrica Bianchi (University of York, York, UK) for the plasmids encoding for human and hamster JUNO, Kazuo Yamagata (Kindai University, Higashiosaka City, Osaka, Japan) for the EGFP-MBD-NLS plasmid, Masahito Ikawa (Osaka University, Osaka, Japan) for the IZUMO1-mCherry transgenic mouse line, and Zene Matsuda (University of Tokyo, Japan) for the DSP plasmids. Finally, we thank Dan Cassel (Technion- Israel Institute of Technology), Leonid Chernomordik (NICHD/DIR, NIH, US), and the members of Podbilewicz Lab for critically reading the manuscript. This project has received funding from the Israel Science Foundation (257/17, 2462/18, 2327/19, and 178/20 to B Podbilewicz) and the European Union's Horizon 2020 research and innovation program under the Marie Skłodowska-Curie Actions (844807 to N G Brukman).

## Additional information

### Competing interests

Nicolas G Brukman, Clari Valansi, Benjamin Podbilewicz: is an inventor on a patent application filed by the Technion- Israel Institute of Technology (US Provisional Patent Application No.532 63/466748), based on this work.

### Funding

| Funder | Grant reference number | Author |
| --- | --- | --- |
| Israel Science Foundation | 257/17 | Benjamin Podbilewicz |
| Israel Science Foundation | 2462/18 | Benjamin Podbilewicz |
| Israel Science Foundation | 2327/19 | Benjamin Podbilewicz |
| Israel Science Foundation | 178/20 | Benjamin Podbilewicz |
| Horizon 2020 Framework Programme | 844807 | Nicolas G Brukman |
| Universidad Nacional Autonoma de Mexico, Direccion General de Asuntos del Personal Academico, Programa de Estancias de Investigacion (PREI), UNAM | | Benjamin Podbilewicz |

The funders had no role in study design, data collection and interpretation, or the decision to submit the work for publication.

### Author contributions

Nicolas G Brukman, Conceptualization, Resources, Data curation, Formal analysis, Funding acquisition, Validation, Investigation, Visualization, Methodology, Writing – original draft; Clari Valansi, Data curation, Formal analysis, Investigation, Visualization, Methodology, Writing – review and editing; Benjamin Podbilewicz, Conceptualization, Supervision, Funding acquisition, Project administration, Writing – review and editing

### Author ORCIDs

Nicolas G Brukman ⓘ https://orcid.org/0000-0002-9865-2503
Benjamin Podbilewicz ⓘ https://orcid.org/0000-0002-0411-4182

## Ethics

All animal studies were approved by the Committee on the Ethics of Animal Experiments of the Technion - Israel Institute of Technology.

## Decision letter and Author response

Decision letter https://doi.org/10.7554/eLife.94228.sa1
Author response https://doi.org/10.7554/eLife.94228.sa2

---

# Additional files

## Supplementary files

• MDAR checklist

• Source data 1. Table containing the raw data from correlation assays (see *Figure 3*).

• Source data 2. Numerical data for all the graphs in:*Figures 1–5*, *Figure 1—figure supplement 1*, *Figure 2—figure supplement 1*, *Figure 3—figure supplement 1*, *Figure 5—figure supplement 1*.

• Source data 3. Raw microscopy images in CZI and TIF formats.

## Data availability

The raw data for the correlation analysis is found in *Source data 1*. In all figures individual data is plotted. The raw numerical data is found in *Source data 2*, the file contains the numerical data for all the graphs in: *Figures 1–5*, *Figure 1—figure supplement 1*, *Figure 2—figure supplement 1*, *Figure 3—figure supplement 1*, *Figure 5—figure supplement 1*. Raw microscopy images in CZI and TIF formats are available in *Source data 3*. Any additional information required to reanalyze the data reported in this paper is available from the lead contact upon request.

---

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
