## [Editor Report]

The molecular mechanisms of mammalian gamete fusion are still obscure. In this study, the authors provide compelling evidence that binding of sperm to somatic cells expressing egg counterpart proteins enables non-reproductive cells to efficiently undergo cell-cell fusion and form a syncytium. This is an important finding that establishes a new experimental system to investigate the proteins involved in egg-sperm binding, and could accelerate innovation of male infertility diagnostic assays.

---

## [Decision Letter]

[Editors' note: this paper was reviewed by Review Commons.]

---

## [Author Response]

General Statements

We are pleased to enclose our revised manuscript entitled “Sperm induction of somatic cell-cell fusion as a novel functional test” for a research article. In this work, we present evidence for a new phenomenon, named SPICER, of sperm-induced fusion of cells expressing the oocyte protein JUNO. We showed that this effect depends on JUNO, the sperm fertilizing ability, and the action of the sperm protein IZUMO1. We were glad to receive positive feedback from the reviewers, in which they understood the potential research and applied applications for our discovery that will be of interest to a wide audience. We have revised and improved the manuscript based on their suggestions.

The main concern that arose from the reviewers was that the applicability of the results to the diagnosis of human infertility is not direct and immediate. In the revised version, we are more cautious in these statements even though we have included new data showing that mouse sperm fertilizing ability correlates with the extent of cell-cell fusion observed (Figure 3C-D).

By request of the reviewers, we have performed an experiment to distinguish the mechanism by which sperm induce syncytia formation (Figure 2—figure supplement 1). When sperm were allowed to fuse to one population of cells before adding the second one, we were not able to detect multinucleated cells with both markers. These results indicate that the simultaneous fusion of one sperm to two adjacent cells is the main mechanism for SPICER to occur.

In addition, we have new controls showing that sperm incubated without calcium or BSA (Figure 3—figure supplement 1), two conditions employed to test the relevance of sperm capacitation, presented lower levels of acrosome reaction (i.e. a functional consequence of capacitation). Also as requested, we carried out the SPICER assay using the hamster JUNO which showed no differences compared to the mouse ortholog (Figure 4). This verifies the functional relevance of the assay by confirming the promiscuous nature of hamster oocytes during in vitro fertilization assays using sperm from different species including humans and mice.

Finally, we added information regarding the number of sperm per syncytia (Figure 1—figure supplement 1A), a concentration curve of sperm (Figure 1—figure supplement 1B), and a time-course experiment (Figure 5—figure supplement 1).

In summary, in this revised manuscript we have carefully considered all the suggestions made by the three reviewers and we enclose a detailed point-by-point response. We have deposited the revised manuscript in BioRxiv.

Point-by-point description of the revisionsReviewer #1 (Evidence, reproducibility and clarity (Required)):This work by Bruckman et al. showed how sperm can induce somatic cell syncytium upon binding. This finding has tremendous implications for the field. Although gamete fusion is one of the fundamental events in biology, the molecular details of this process in mammals is not well understood. As briefly mentioned in this work, spermatozoa contribute Izumo1 which forms a complex with SPACA6 and TMEM81; and probably with other proteins. On the other hand, the egg counterpart is JUNO, a GPI- anchored protein, and CD9. IZUMO1 and JUNO have been shown to interact and this interaction is essential for fusion. However, it is still not clear the extent that this binding is needed for attachment of for fusion between gametes. The authors have published before that somatic cells expressing IZUMO1 can fuse with other cells expressing JUNO; and that sperm can fuse to JUNO-expressing cells. This previous work strongly suggests that IZUMO and JUNO are sufficient for fusion; however, the efficiency of the fusion assay was low. In the present work, formation of syncytium is highly efficient, making possible to envision applications of this assay to research of the proteins involved in sperm-egg fusion as well as for molecular mechanisms. In addition, as mentioned in this work, the assay can be translated for the use for clinical diagnosis.

We would like to thank the reviewer for the insightful comments that improved our manuscript. Furthermore, we appreciate the acknowledgment of the value of our findings for the field.

Overall, the manuscript is straightforward, and the conclusions are sound. I have only a couple of comments:1) As mentioned in this work, fusion of human sperm with hamster eggs was used in the past to evaluate human sperm fusogenic properties. Although this assay is not longer recommended, still highlight the fact that hamster egg's oolemma is promiscuous regarding species-specific fusion. Although I don't think the experiment is essential, if it were possible, trying hamster IZUMO1 in this assay will be a good addition to this manuscript.

We have performed this experiment and it is now included in the revised manuscript (Figure 4). As discussed in the new version, hamster JUNO is as efficient as its murine counterpart in facilitating SPICER, confirming the well-known promiscuous nature of hamster eggs.

2) Is it possible to know how many sperm form part of the syncytium>?

We have included in the revised manuscript a Figure showing the number of sperm fused in syncytia of different sizes (Figure 1—figure supplement 1).

3) Although I understand the relevance of this assay for diagnostic use, still the major contribution of this manuscript is to the sperm-egg fusion field of study. I encourage the authors to start the discussion with a more basic approach and add at least one paragraph summarizing how their discovery advanced the field. From my perspective the translational application should be at the end of the Discussion section, not at the start.

We have modified the Introduction and Discussion sections according to this comment. In addition, we are now including new data further supporting that the SPICER assay can be used to predict sperm fertilizing capacity (Figure 3).

Referee Cross-CommentingThe syncytia described in this work support the hypothesis that once Izumo and partners in sperm interact with Juno, the cells become able to undergo fusion events. It does not claim that this type of event happens in vivo. Although Juno is expressed in other cell types and not only in the egg, it is too soon to claim a physiological role for syncytia formation in vivo.I agree with reviewer 2 comments. Both experiments proposed (paragraph 2 and 3) are relatively straightforward.In general, I also agree with reviewer 3 comments, one of the experiments proposed is similar to the one proposed by reviewer 2. Having said this, I disagree that the human sperm experiment is needed for this paper to be significant. Although the diagnostic claim is important, the major contribution of this work is at the basic research level. Toning down the clinical relevance would be sufficient to make this paper a significant contribution to the field.Mechanisms of sperm-egg fusion have been elusive for the last decades. In my view, syncytia formation and I don't expect a single paper to elucidate such a complex event.Reviewer #1 (Significance (Required)):General Assessment: This is an excellent manuscript contributing to our understanding of sperm-egg fusion. In addition, the tools described here warrant new approaches to study the molecules involved in this process using genetically modified mouse models.Advance: Although the relevance of IZUMO1 and JUNO for sperm-egg fusion have been previously reported, as far as I know, this is the first study showing somatic cell fusion and syncitium formation using sperm and somatic cells. This finding will provide new tools to study the molecular basis of sperm-egg fusion as well as generate translational tools to evaluate human sperm ability to fertilize.Audience: I believe that this manuscript will have broad interest. Although initially, the main audience will be related to reproductive biologists, this manuscript will be also highly relevant for other scientists studying fusion. In addition, this work has clear translational applications.Keywords describing my expertise: sperm, eggs, in vitro fertilization, reproductive biology, embryos

We are pleased and grateful to read this reviewer’s comments. We have now implemented the suggestions in the revised version.

Reviewer #2 (Evidence, reproducibility and clarity (Required)):This is a very nice manuscript that continues previous work of the lab. Authors in this manuscript goes further, showing that proteins from the plasma membrane of sperm end up in somatic cells plasma membrane after fusing, conferring them fusogenic capacity. The manuscript is very elegantly written, and of interest to the reproductive community. I have only some minor concerns to be addressed:

We would like to thank the reviewer for the positive feedback as well as for the valuable suggestions.

1. The authors indicate that this test "can help embryologists to better predict the success of the different ARTs for each individual.". This assumption should be taken as it is, an assumption. In order to state that in can be used, authors should first demonstrate, maybe with a correlation test hand by hand with a reproductive clinic, that this procedure indeed works for diagnosis. There is no doubt that it is a very nice proof of principle.

We have changed the Introduction and Discussion to be more cautious about this point. Unfortunately, we currently do not have permission to work with human samples, however, we have added new data showing a positive correlation between the extent of cell-cell fusion and the fertilizing potential of mouse sperm (Figure 3).

2. Figure S1C. Although I agree with the ayuthors that the two models are possible, have the authors tried to wash sperm away from the fusing assay, and add new cells, maybe differentially tagged, to see whether they fuse to those already fused with sperm? This could shed light onto the mechanism.

We have performed this experiment that was requested by Reviewer #1 and Reviewer #2 (Figure 2—figure supplement 1). The results show that when the sperm are added to the plate before the addition of the second population of cells, no fusion is observed. This argues against the model in which there is transfer of the fusion machinery to the somatic cells. We have now adjusted the manuscript accordingly.

3. Figure 3. Sperm incubated in the absence of BSA, failed tu fuse to cells. I wonder whether the lack of BSA precludes fusogenicity. Have authors tried removing HCO3 and keeping BSA? The presence of HCO3 still enables PKA activation even in the absence of BSA, also allowing hyperpolarization of the plasma membrane. Have authors evaluated acrosomal status after the selected treatments?

As the co-incubation of the sperm and the cells is performed in the cell-culture incubator with CO_2_ supply, we were not able to evaluate a condition without HCO3. However, we have analyzed the levels of acrosome reaction after the incubation in media without calcium or BSA (Figure 3—figure supplement 1). For both conditions, the levels of acrosome reaction are significantly lower than the control consistent with previous reports (Visconti et al., 1995 PMID: 7743926).

Referee Cross-CommentingI thank reviewer 3 for his/her insight. I agree on being cautious about the relevance of this finding, as it is unclear if anything similar happens in vivo.I agree with comments raised by rev 1. This reviewer clearly points out that the clinical relevance of the manuscript is not a strength of the manuscript. And I agree with reorganizing the discussion as suggested. In this line, I consider very important to tone down the clinical aspect of the manuscript. In this regard, I completely understand the experiments suggested by reviewer 3, but still, I think that the major contribution of this manuscript is to the sperm-egg fusion field of study. Thus, my suggestion is to tone down to a minimum level the clinical relevance of the study.Reviewer #2 (Significance (Required)):The work shows an interesting advance in the field. However, caution should be taken when referring to "capacitated sperm", since no controls are taken, and methods to inhibit capacitation are not solid.

We thank the reviewer for this comment. We have improved the manuscript following their recommendations and now the pertinent controls are included (Figure 3—figure supplement 1).

Reviewer #3 (Evidence, reproducibility and clarity (Required)):Summary:The authors find that mouse sperm can fuse to hamster fibroblasts (Baby Hamster Kidney, BHK) and that consequently these cells form large multinucleated syncytia. They describe and quantify the formation of syncytia and suggest that the sperm ability to induce formation of syncytia is associated with sperm functionality.Based on these findings, the authors propose a novel method for the diagnosis of male infertility.Overall, the manuscript is clearly written, and the experimental procedures are described in detail.

We sincerely appreciate the important comments and suggestions made by Reviewer #3.

Major comments:The main limitation of the study is that the data is obtained with mouse sperm, while the proposed application is for the evaluation of human sperm fertilizing ability. The authors need to repeat the assay of sperm fusion to somatic cells with human samples.Furthermore, to correlate sperm functionality with cell fusion induction, it would be necessary to compare sperm from fertile and infertile men. The experiment reported in figure 3, where non-capacitated sperm, maintained in media without Calcium or without BSA, are used, does not mimic real cases of human male infertility where even fully capacitated sperm are unable to fertilize eggs.The experiments with human sperm should not take longer than 6 months, depending on the local legislation regulating human sample collection and usage.

We understand the point made by Reviewer #3 and agree to tone down the clinical significance of the finding. Unfortunately, we do not have permission to perform experiments with human samples. We also agree with Reviewers #1 and #2 in the sense that these experiments are out of the scope of this study. However, we have now included new data directly correlating mouse sperm fertilizing ability and syncytia formation (Figure 3) that further support our conclusions.

Minor comments:Figure 1. The % of multinucleation in panel C (~ 55%) and in panel D (25%) are substantially different. The authors should explain better how the percentages were calculated.

As explained in the Results section, within the panels and in the legend of Figure 1, in panel C we quantify multinucleation separately for cells with and without sperm fused to them to distinguish the effect of sperm fusion. Instead, in panel D we quantify multinucleation in the whole population. This is better clarified in the revised version in the legend of Figure 1.

Figure 2., panel C. It is unclear how the number of 'fluorescent cells IN CONTACT that do not fuse (NuC)' are evaluated given the lack of a marker for cell membranes or for actin. It is challenging to visualize all the filopodia at the edges of cells just with light microscopy, therefore adding a cell membrane dye could be helpful for visualization and counting.

We agree with the reviewer that we cannot confirm that the cells are projecting filopodia between them. When we stated “in contact” we referred to the cell body, as we have done in several publications to date: Podbilewicz et al., 2006 (DOI 10.1016/j.devcel.2006.09.004); Avinoam et al., 2011 (DOI: 10.1126/science.1202333); Valansi et al., 2017 (DOI: 10.1083/jcb.201610093); Moi et al., 2022 (DOI: 10.1038/s41467-022-31564-1); and Brukman et al., 2023 (DOI: 10.1083/jcb.202207147). To be more specific we have now explained this in the revised version as “cells whose cell bodies are in contact”.

Figure 3, panel A. The authors should specify if the 500 cells are single cells or nuclei, including those in syncytia.

The figure refers to cells, not nuclei. It is better explained in the revised version of the Materials and methods.

Figure S1 panel C. To discern between the mechanism proposed in (i) and (ii), a possibility would be to add sperm to JUNO(GFPnes) cells, wash them out carefully after 4 hours in order to remove non-fused and unbound sperm, and then to add JUNO(H2B-RFP) cells. If sperm induce multinucleation by fusing with more than one cell, only GFP syncytia will be obtained. On the other hand, if fusion is due to the transfer of sperm fusing machinery, syncytia will be GFP positive and have red nuclei too.

We performed this important experiment that was also requested by Reviewers #1 and #2 (Figure 2—figure supplement 1). We found that when the sperm were added to the plate before the addition of the second population of cells, no hybrid syncytia were formed, only multinucleated GFP cells. The manuscript has been modified to include this observation that points against the model in which there is transfer of the fusion machinery to the somatic cells.

Referee Cross-commentingAs pointed out by reviewer 1 'this is the first study showing somatic cell fusion and syncitium formation using sperm and somatic cells' therefore if the clinical relevance of the manuscript is toned down, I believe that the authors need to add a more in-depth investigation of the cell fusion mechanism. Particularly in consideration of the data shown by other groups indicating that IZUMO1 and JUNO are not sufficient for cell fusion. In that regard, the experiments suggested by reviewers 1 and 2 would be helpful to start getting a better understanding of the mechanism. Anyhow, I would be extremely cautious about the relevance of this finding because it is unclear whether anything similar happens in vivo.Reviewer #3 (Significance (Required)):The mechanism of sperm-egg cell membrane fusion is still unclear, and the entire molecular machinery orchestrating fertilization is yet to be unveiled.The sperm protein IZUMO1 and its egg receptor JUNO have been shown to be essential for sperm-egg binding in mouse, human and rats. Whether IZUMO1 also mediates membrane fusion is still debated because research groups who have adopted various assays came to different conclusions (PMID: 24739963; PMID: 35096839; PMID: 36394541; PMID: 26568141).In a previous paper published earlier this year Brukman and colleagues showed that IZUMO1 is able to induce membrane fusion in a heterologous cell system (PMID: 36394541). In the current manuscript, they expand on this observation and report the formation of syncytia (large multinucleated cells) induced by the fusion of sperm with somatic cells. Whether this event that is observed in vitro has any relevance in vivo has to be investigated.The large amount of work done on an unrelated cell membrane protein, the fibroblast growth factor receptor-like 1 (FGRL1), that also induces cell fusion in vitro, suggests a possible interpretation for the results described by Brukman et al. 'Cell fusion might just be the most extreme result of very tight cell adhesion […] Under normal conditions, FgfrL1 might only bring together the cell membranes into intimate contact' https://link.springer.com/article/10.1007/s00018-012-1189-9 (PMID:23112089)Therefore, it is reasonable to think that cell-cell fusion observed in vitro does not recapitulate the fusion of sperm and egg.Regardless of whether sperm can induce fusion of somatic cells in vivo, the SPICER method proposed in this manuscript needs to be validated with human sperm in order to become a valuable tool for the assessment of male fertility.

We are truly grateful for the comments and suggestions made by Reviewer #3 that allowed us to improve our manuscript. We agree about the important questions that arise from our discovery and we believe that SPICER will be a powerful tool to address them in the future. We would only like to mention that the FGRL1-induced fusion was observed only when at least one of the fusing cells was a CHO cell (PMCID: PMC2988375 DOI: 10.1074/jbc.M110.140517). In our case, the fact that SPICER, as well as IZUMO1-induced fusion (Brukman et al., 2023, J Cell Biol.), was detected in two unrelated cell lines (i.e. hamster BHK and human HEK cells) argues against a cell-specific effect and suggests the involvement of a conserved mechanism.